# A non-neutralizing antibody broadly protects against influenza virus infection by engaging effector cells

Yi-An Ko☉, Yueh-Hsiang Yu☉, Yen-Fei Wu, Yung-Chieh Tseng, Chia-Lin Chen¤a, King-Siang Goh, Hsin-Yu Liao, Ting-Hua Chen¤b, Ting-Jen Rachel Cheng, An-Suei Yang, Chi-Huey Wong, Che Ma‡, Kuo-I Lin‡*

Genomics Research Center, Academia Sinica, Taipei, Taiwan

☉ These authors contributed equally to this work.
¤a Current address: Merck Research Laboratory, Merck Sharp & Dohme (I.A.) LLC, Taiwan Branch, Taipei, Taiwan
¤b Current address: Okinawa Institute of Science and Technology Graduate University, Onna, Okinawa, Japan
‡ CM and K-IL share senior authorship on this work.
* kuoilin@gate.sinica.edu.tw

**Data Availability Statement:** All relevant data are within the manuscript and its Supporting Information files.

## Abstract

Hemagglutinin (HA) is the immunodominant protein of the influenza virus. We previously showed that mice injected with a monoglycosylated influenza A HA (HA$_{mg}$) produced cross-strain-reactive antibodies and were better protected than mice injected with a fully glycosylated HA (HA$_{fg}$) during lethal dose challenge. We employed a single B-cell screening platform to isolate the cross-protective monoclonal antibody (mAb) 651 from mice immunized with the HA$_{mg}$ of A/Brisbane/59/2007 (H1N1) influenza virus (Bris/07). The mAb 651 recognized the head domain of a broad spectrum of HAs from groups 1 and 2 influenza A viruses and offered prophylactic and therapeutic efficacy against A/California/07/2009 (H1N1) (Cal/09) and Bris/07 infections in mice. The antibody did not possess neutralizing activity; however, antibody-dependent cellular cytotoxicity and antibody-dependent cellular phagocytosis mediated by natural killer cells and alveolar macrophages were important in the protective efficacy of mAb 651. Together, this study highlighted the significance of effector functions for non-neutralizing antibodies to exhibit protection against influenza virus infection.

## Author summary

The protective efficacy of antibodies is generally related to their neutralization potency. Here, we isolated a monoclonal antibody from mice injected with monoglycosylated hemagglutinin protein-based universal influenza vaccine, and demonstrated a head-domain recognizing, but non-neutralizing, monoclonal antibody carried prophylactic and therapeutic efficacy against a broad spectrum of influenza virus infections *in vivo* via effector functions.

**Funding:** This work was supported by grants from Academia Sinica (AS-SUMMIT-109 to CHW, AS-IA-107-L05 to KIL; https://www.sinica.edu.tw/en), Taiwan, and Ministry of Science and Technology (MOST 109-2320-B-001-023-MY3 to KIL; https://www.most.gov.tw/?l=en). The funders had no role in study design, data collection and analysis, decision to publish, or preparation of the manuscript.

## Introduction

Influenza viral infections cause a contagious respiratory illness of the upper airways and lungs. Approximately half a million deaths worldwide are due to seasonal influenza each year [1]. Influenza viruses belong to the *Orthomyxoviridae* family, whose genomes encode 14 proteins, including the major surface glycoproteins hemagglutinin (HA) and neuraminidase (NA) [2]. Influenza viruses are classified into four subtypes: A, B, C, and D. Among which, the A and B subtypes are responsible for the seasonal influenza in humans [2]. Although the influenza vaccine is currently in use for controlling the spread of seasonal influenza each year, the trivalent inactivated influenza vaccine confers variable degrees of protection in those vaccinated individuals depending on the extent of mismatch between vaccine strains and influenza virus strains circulating in the population [3]. More severe complications resulting from influenza viral infection and a loss of influenza vaccine efficacy have been found in older adults [4,5]. Thus, there remains a need to develop new strategies to overcome the moderate responses and narrow coverage range of the current flu vaccine.

Advances in antibody engineering technologies have enabled the isolation of several neutralizing antibodies against influenza viruses from either infected patients or donors with extensive vaccinations [6–10]. These anti-influenza neutralizing monoclonal antibodies (mAbs) target HA, which comprises a variable immunodominant globular head domain or a more conserved immune-subdominant stem domain [11,12]. Most of the broadly neutralizing antibodies target the stem region of HA and have less potent direct neutralization activity [11–13]. Other neutralizing antibodies bind to the head region of HA and generally possess a potent ability to directly inhibit virus entry [12]. Recently, non-neutralizing mAbs recognizing the HA globular head domain trimer interface have been reported that limit the spread of influenza viruses and protect against infection by various influenza virus strains in mice [10,14]. In addition to the abovementioned modes of action, antibodies can eliminate infected cells through effector functions, such as antibody-dependent cellular cytotoxicity (ADCC) [15], in which natural killer (NK) cells with Fc receptors (FcRs) are the primary effector cells [16]. It has been demonstrated that mAbs targeting the stem region of HA can mediate ADCC [17]. Additionally, antibody-dependent cellular phagocytosis (ADCP), which involves the ingestion of antibody-opsonized influenza virus particles by alveolar macrophages, has also been shown to be involved in the clearance of influenza viruses [18,19].

Changing the glycan abundance of HA can affect its immunogenicity [20–22]. We developed a monoglycosylated HA ($HA_{mg}$) protein-based universal influenza vaccine by treating HA with endoglycosidase H. $HA_{mg}$ served as an effective vaccine providing broader protection against infection by various influenza virus strains than the fully glycosylated HA ($HA_{fg}$) in animal models [23,24]. One mAb, 651, isolated from $HA_{mg}$-immunized mice, was subjected to functional characterization in this study. Although it lacks neutralizing activities *in vitro*, 651 is able to recognize the head region of a broad range of HA proteins and protect against infection by H1N1 viruses through effector functions *in vivo*. Thus, we demonstrated a unique mode of action for a non-neutralizing mAb that can offer cross-strain protection against influenza virus infection through FcR-mediated effector functions. The results highlight the potential significance of non-neutralizing antibodies in host responses against viral infections.

## Results and discussion

### Generation of mAb 651 from Bris/07 $HA_{mg}$-immunized mice

We previously demonstrated that the $HA_{mg}$ vaccine elicits antibody responses that recognize a broader spectrum of influenza viruses than the $HA_{fg}$ vaccine [24]. We aimed to isolate broadly

neutralizing mAbs from mice immunized with $HA_{mg}$ proteins prepared from the A/Brisbane/ 59/2007 (H1N1) (Bris/07) influenza virus. By using a single B-cell antibody screening technique, we isolated and expressed the $HA_{mg}$-specific chimeric mAbs (Fig 1A). We identified a specific mAb, 651, which recognized the broad spectrum of HA proteins from different strains of influenza virus, including group 1 and group 2 influenza A virus subtypes (Fig 1B). Quantitative measurement using an antibody affinity assay showed that 651 possessed a higher affinity to $HA_{mg}$ than $HA_{fg}$ from both A/Brisbane/59/2007 (H1N1) (Bris/07) and A/California/7/ 09 (H1N1) (Cal/09) (Fig 1C).

The structure and conformation of HA are important for its recognition by 651, as decreased binding was observed after HA was treated with 2-ME, which disrupted its protein conformation (Fig 1D). ELISA results showed that 651 bound to the intact HA, but not to the stem region, of Bris/07 (Fig 1E), implying that 651 mainly recognizes the globular head domain of HA. Supporting this notion, the results of a hydrogen-deuterium exchange-mass spectrometry (HDX-MS) assay showed that there was one region recognized by 651. The mAb 651 interacts with the globular head region distinct from the receptor-binding site and near a glycosylation site of HA from Bris/07 (S1A Fig). Sequence alignment of the identified Bris/07 HA peptide with the HA proteins from several influenza viruses revealed that this 651-binding region in the head globular domain is relatively conserved in Bris/07 and Cal/09, and is close to the N-glycosylation site (S1B Fig). To further validate the HA epitope that is recognized by 651, we performed competitive ELISA by using two mAbs, C05 and F10, that bind to the receptor-binding site of head region and stem region of HA, respectively [7,25] (S1C Fig). Indeed, both mAbs failed to compete the binding of 651 with HA (S1D Fig), confirming the HDX-MS results showing that the potential binding site of 651 is distinct from the receptor-binding site on head region.

Typically, anti-influenza virus antibodies that recognize the globular head domain possess hemagglutination inhibition (HI) and neutralizing activity but show less cross-reactivity compared with antibodies against the stem region [12,26]. Although 651 recognizes the HA of a panel of influenza viruses, it does not possess neutralizing activity against Bris/07 (H1), Cal/09 (H1), NIBRG-14 (H5), or Vic/11 (H3) influenza viruses (Fig 1F). Furthermore, 651 did not show HI activity against the Cal/09 virus (S1E Fig). Therefore, we have isolated a mAb 651 derived from Bris/07 $HA_{mg}$-immunized mice that targets the globular head region of various HA proteins but has no neutralizing activity.

## mAb 651 confers preventive and therapeutic efficacy through FcγR

Because 651 binds to various strains of influenza virus, we next sought to understand its biological functions *in vivo*. Mice were intraperitoneally injected with two different 651 doses 2 h before viral infection (Fig 2A). To our surprise, pretreatment with 651 provided a dose-dependent protection against Bris/07 infection (Fig 2B) and a complete protection against Cal/09 infection (Fig 2C). The better prophylactic efficacy against Cal/09 correlated with the stronger binding affinity of 651 to the HA of the Cal/09 strain than to that of Bris/07 as the $K_D$ between 651 and Cal/09 HA was lower than that between 651 and Bris/07 (Fig 1C).

We then examined the therapeutic efficacy of 651. Mice were administered the antibodies 1 day after infection with influenza virus (Fig 2D). Again, the survival rates were significantly improved in the 651-treated groups after Cal/09 infection. Improved, although not statistically significant, survival rates were observed in the high-dose 651-treated mice after Bris/07 infection (Fig 2E). Remarkably, mice provided a high dose of 651 1 day after Cal/09 infection were completely protected from a lethal dose Cal/09 challenge on day 14 (Fig 2F). Weight loss following viral infection was also prevented in the mice treated with 651 before or after infection

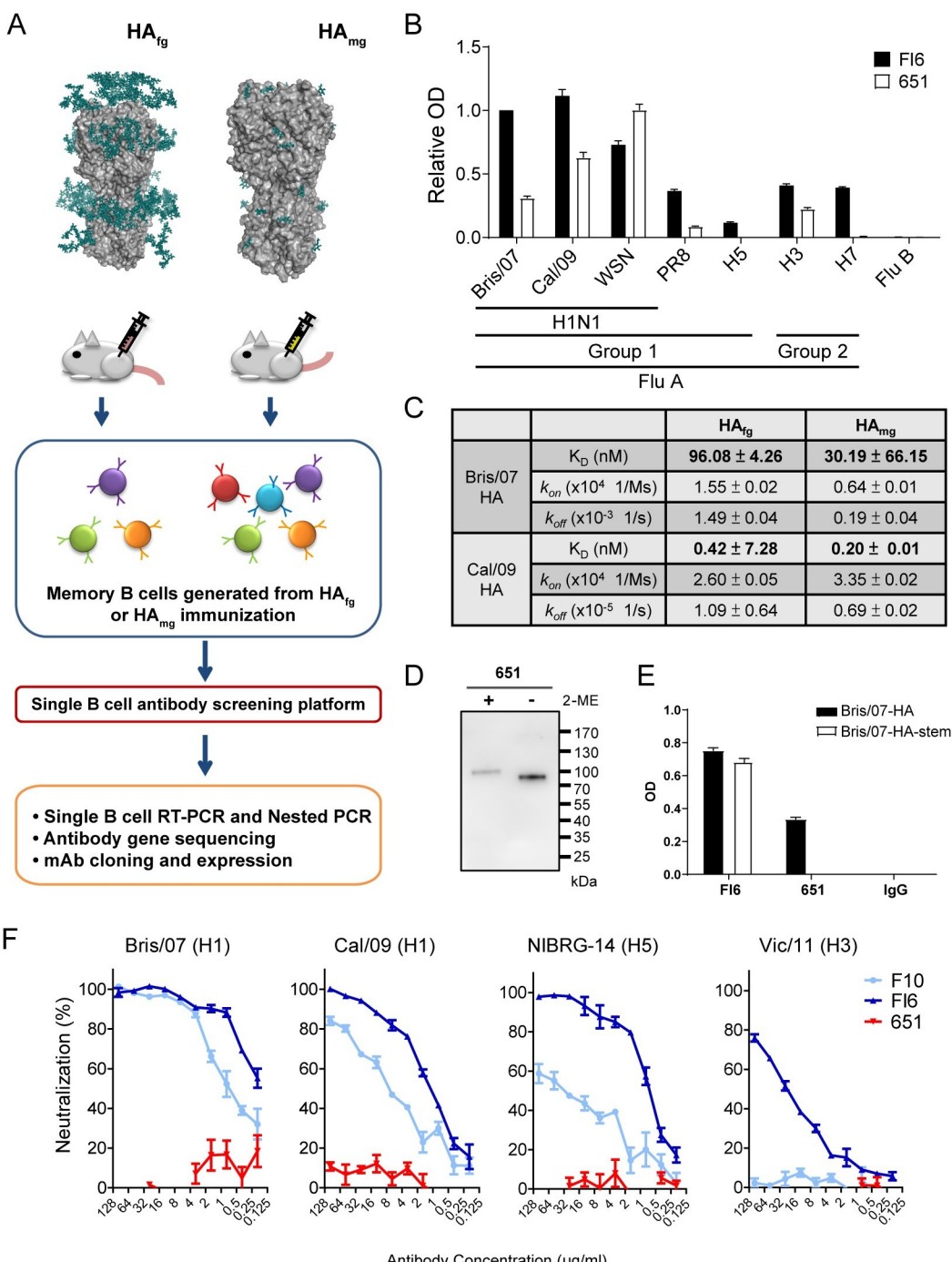

**Fig 1. Generation of 651 from Bris/07 HA$_{mg}$-immunized mice.** (A) Bris/07 (H1N1) HA proteins with two types of glycan modifications, HA$_{fg}$ (HA carrying the typical complex type N-glycans) and HA$_{mg}$ (HA carrying GlcNAc at N-glycosylation sites only), were used to immunize mice, followed by isolation of mAbs with a single B cell screening platform. (B) The resulting recombinant mAb 651 showed cross-recognition of several strains of group A influenza virus. (C) BLI assay showing 651 possessed a higher affinity for HA$_{mg}$ from Cal/09. (D) Immunoblot of reduced (5% 2-ME in sample buffer) and non-reduced Bris/07 HA proteins showing 651 recognized the structural epitope of HA. (E) ELISA showing 651 was unable to recognize the stem region of HA from Bris/07. (F) Effects of 651 on the microneutralization of various strains of influenza viral infections. FI6 and F10 were used as the positive controls. Data are shown as the mean ± SEM (n = 3).

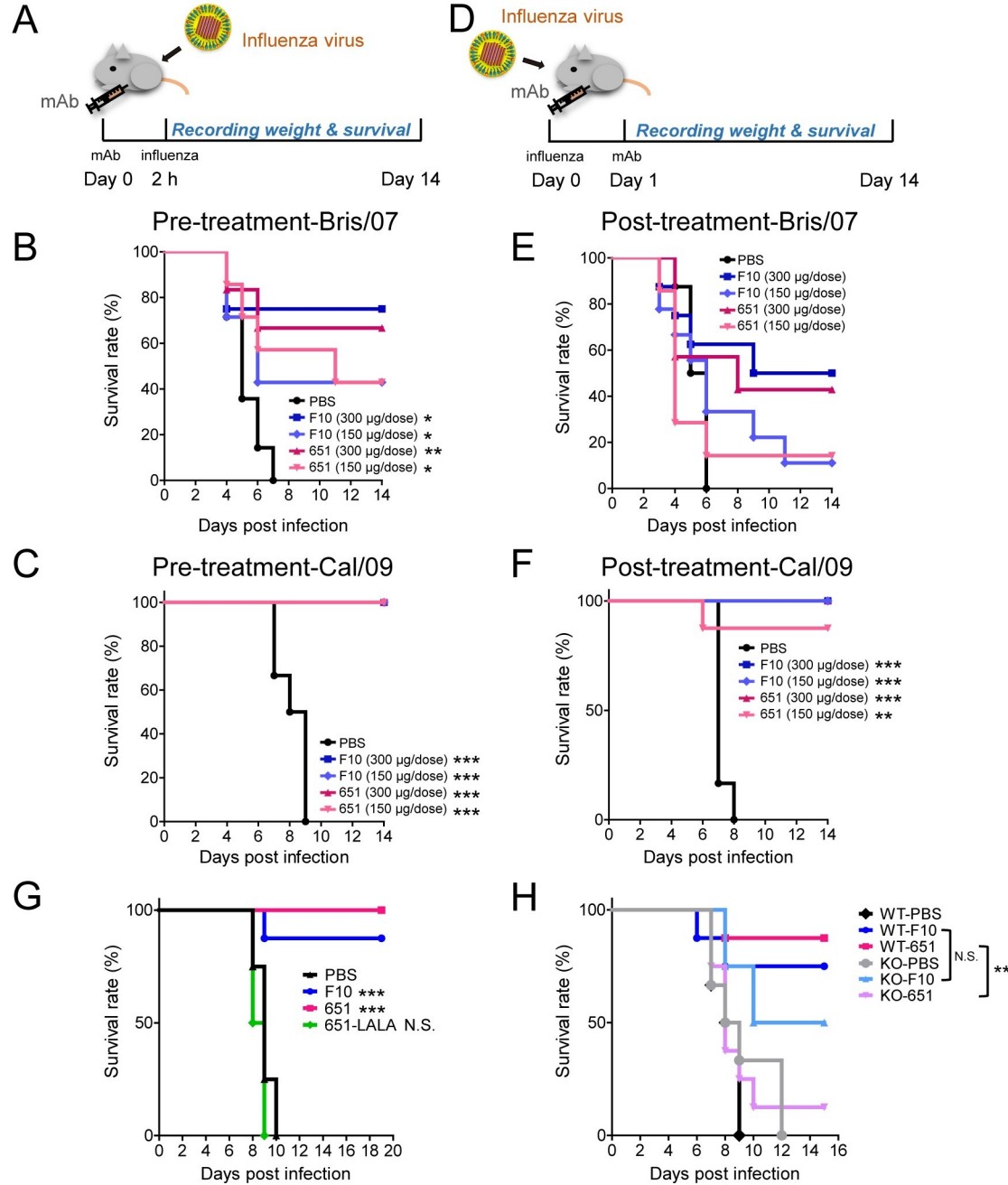

**Fig 2. Prophylactic and therapeutic efficacy of 651 in mice challenged with lethal dose of H1N1.** (A-C) Cumulative survival rate of mice treated (i.p.) with indicated mAb 2 h before challenge with 100 LD50 of H1N1 Bris/07 virus (B) or Cal/09 virus (C). n = 4–14 mice per group in B and 6–8 mice per group in C. (D-F) Cumulative survival rate of mice injected (i.p.) with 651 24 h after challenge with 100 LD50 of H1N1 Bris/07 virus (E) or Cal/09 virus (F). n = 7–9 mice per group in E and 6–8 mice per group in F. (G) Cumulative survival rate of mice treated with F10, 651, or a mutation of 651 (651-LALA) after challenge with Cal/09 at 100 LD50. n = 8 mice per group. (H) Cumulative survival rate of FcγR knockout (KO) mice and control WT mice after F10 or 651 pretreatment and Cal/09 virus challenge. n = 6–8 mice per group. Data were analyzed by log-rank (Mantel-Cox) test. The statistical significance was calculated for the differences between PBS control and mAb treated groups (in B, C, E, F, and G), or between WT and FcγR knockout (KO) mice receiving the same kind of mAb (in H). *p < 0.05; **p < 0.01; ***p < 0.001. N.S. = no significant difference.

(S1A–S1D Fig). F10 was used as a positive control antibody, as it has prophylactic and therapeutic efficacy against a broad spectrum of group 1 human influenza A viruses [25]. Our results showed 651 and F10 had comparable preventive and therapeutic efficacies (Figs 2A–2F and S2A–S2D). Thus, 651 provided notable protective efficacy against influenza viral infection *in vivo*, despite being a non-neutralizing antibody.

In addition to being able to directly neutralize pathogens, antibodies can eliminate infection through FcR-mediated reactions, such as ADCC or ADCP [12]. Because 651 was unable to neutralize the influenza virus, we next examined whether the protective efficacy of 651 was due to FcR-mediated effector functions. We generated an Fc-region mutant with leucine 234 and 235 to alanine substitutions, called 651-LALA, which mitigates antibody effector function by abolishing antibody binding to the FcR [27]. We found that mice given 651-LALA had similar mortality rates as those pretreated with PBS (Fig 2G), demonstrating the significance of the effector function of 651. The γ chain of the Fc receptor is a critical component of the high-affinity receptor for IgG [16]. Mice lacking the FcR γ chain displayed impaired NK-cell-mediated ADCC and macrophage-mediated ADCP [28]. Significantly reduced survival rates were also found in 651-pretreated *Fcer1g*-knockout mice compared with 651-pretreated wildtype mice after Cal/09 infection (Fig 2H). However, a lack of *Fcer1g* did not abolish the protective effect of F10, indicating that neutralizing activity predominantly contributed to the function of F10 (Fig 2H). Following challenge with a lethal dose of Cal/09 virus, mice pretreated with 651-LALA or *Fcer1g*-knockout mice consistently showed similar dramatic bodyweight loss as the PBS treated group (S2E and S2F Fig). Together these results indicate that the protective effects mediated by 651 rely on FcR-mediated effector functions.

## 651-pretreatment alleviates virus replication and ameliorates lung inflammation

Upon influenza viral infection, both virus-induced virulence and immunopathology contribute to inflammation and tissue injury in the respiratory tract [29]. We next examined whether 651 leads to viral clearance and alleviates lung inflammation. Virus replication was abolished in mice pretreated with F10 2 h before Cal/09 infection by 3 days post infection, which is likely to be due to the potent influenza-virus-neutralizing ability of F10 (Fig 3A). Notably, a significant improvement in viral clearance was seen on day 5, but not day 3, in Cal/09-infected mice pretreated with 651 (Fig 3A and 3B). Hematoxylin and eosin (H&E) staining further showed the reduced pulmonary edema and immune cell infiltration in mice pretreated with F10 or 651. However, mice treated with 651-LALA displayed lung damage and immune cell infiltration as severe as that seen in PBS-treated mice on day 3 after Cal/09 infection (Fig 3C).

The viral nucleic acids released upon influenza virus infection can be sensed by various pattern-recognition receptors, which trigger robust downstream signaling [30,31]. Several cytokines in the lungs, including the type I IFN, MCP-1, IL-6, and TNFα produced by various cells after influenza infection, lead to cytokine storms—one of the major causes of severe flu-associated complications [30]. We next assessed whether the production of cytokines in the lungs is influenced by 651 administration. We found that the production of cytokines, including IFNα, IFNβ, MCP-1, IL-6, and TNFα, was elevated on days 3 and 5 after Cal/09 infection (Fig 3D and 3E). Pretreatment with F10 or 651 significantly reduced the release of proinflammatory cytokines on day 3 (Fig 3D). At 5 days post infection, the production of IFNβ, MCP-1, and IL-6 remained significantly reduced by pretreatment with F10 or 651 (Fig 3E). Type III IFN, IFNλ, has been shown to have therapeutic efficacy against the influenza virus, but it does not induce proinflammatory effects [32]. It is noteworthy that the IFNλ level was elevated in mice pretreated with F10 or 651 on day 3 post infection (Fig 3D). The 651-mediated reduction of

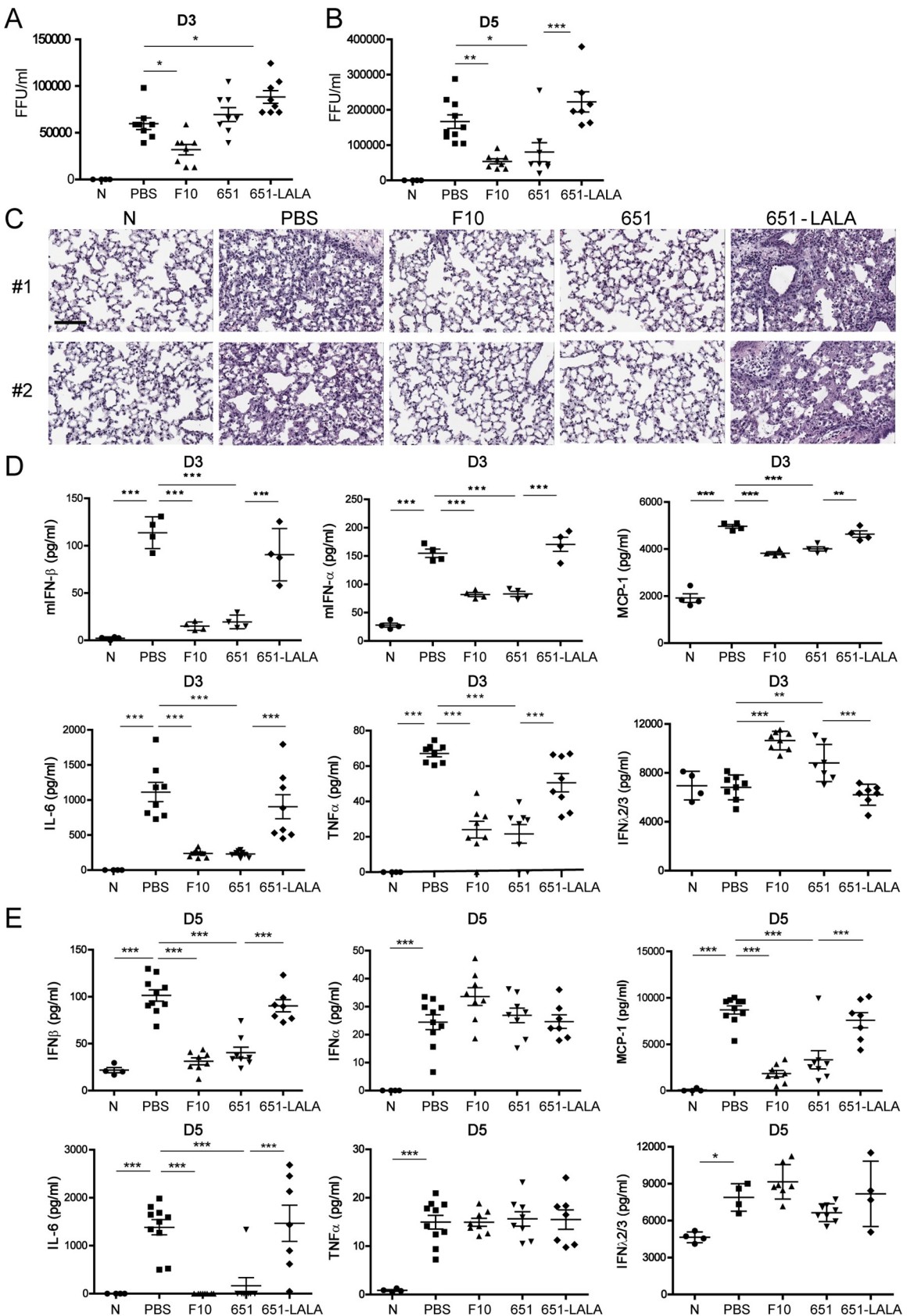

**Fig 3. Pretreatment with 651 reduced viral replication and alleviated lung inflammation after respiratory challenge with Cal/09 virus.** (A and B) Fluorescent focus assay showing the viral titers in lungs of various indicated mAb- or PBS-pretreated mice on day 3 (A) and day 5 (B) after Cal/09 infection. n = 4–8 mice per group in A and 4–10 mice per group in B. (C) H&E staining showing the histopathology of lungs in mAb-pretreated mice on day 3 after Cal/09 infection. Results from two individual mice from each group are shown. Scale bar = 100 μm. (D and E) Pulmonary IFNα, IFNβ, MCP-1, IL-6, TNFα, and IFNλ2/3 protein levels on day 3 (D) and day 5 (E) after Cal/09 infection were measured by ELISA using samples from mice receiving indicated pretreatments. n = 4–8 mice per group in D and 4–10 mice per group in E. N represents naïve mice without infection. Results are mean ± SEM. *p < 0.05; **p < 0.01; ***p < 0.001. Data are shown as the mean ± SEM.

cytokine production was reverted when Leu234/Leu235 sites were mutated (Fig 3D and 3E). Therefore, similarly to pretreatment with F10, pretreatment with 651 alleviated lung inflammation after influenza virus challenge.

## ADCC and ADCP mediated by 651

The HA of influenza virus was reported to be the antigenic determinant responsible for the generation of antibodies with ADCC [33]. Further, anti-neuraminidase (NA) antibodies are also able to elicit weak ADCC and to boost ADCC induced by antibodies against stem region of HA [34]. Several immune cell types, including NK cells, target antibody-labeled cells after infection and release cytotoxic granules and cytokines to kill the virus-infected cells [35]. Following Cal/09 challenge, the percentage of NK cells, defined as CD49b⁺CD3e⁻, in the lungs increased on day 3 (Fig 4A, left panel) but then declined on day 5 (Fig 4A, right panel). We found an increased NK cell number in the lungs on days 3 and 5 after Cal/09 infection (Fig 4B). Furthermore, we used HEK293T cells to express HA proteins from various strains of influenza virus, including Bris/07 H1N1, Cal/09 H1N1, H3N2, and H7N9, to examine whether 651 provided ADCC *in vitro*. Compared with the effects in the control IgG-treated cells, both 651 and FI6 showed greater dose-dependent cytotoxicity towards cells expressing Bris/07, Cal/09, and H3 (Fig 4C), but 651 did not provide ADCC against H7-expressing cells (Fig 4C). This is consistent with our data showing the lack of binding to H7 by 651 (Fig 1B). Since ADCC relies on the Fc-FcR interaction, 651-LALA could not trigger effective ADCC against Bris/07 and Cal/09 HA (Fig 4D).

Besides ADCC, ADCP also contributes to protection against influenza virus infection. Non-neutralizing antibodies have been shown to offer protective effects through alveolar macrophages but not NK cells [19]. We found that on days 3 and 5 after Cal/09 challenge, the frequency of alveolar macrophages significantly declined (Fig 4E and 4F), which is consistent with a previous report showing that alveolar macrophages dramatically diminished in BLAB/c mice after influenza virus infection [36]. Pretreatment with F10 or 651 significantly rescued the decline of alveolar macrophages caused by influenza virus infection on days 3 and 5 post infection (Fig 4E and 4F). We subsequently used THP-1 as effector cells to confirm whether 651 utilizes ADCP for influenza virus clearance *in vitro*. The antibody opsonized Cal/09 virus was incubated with sialidase-treated THP-1 cells, and intracellular viral N protein (NP) was detected. We found that the uptake of viruses by THP-1 cells was accelerated by opsonized 651 and F10 (Figs 4G, S3A and S3B) compared with uptake by the IgG- and 651-LALA-opsonized group. Moreover, a dose-dependent ADCP by 651 was observed (Fig 4H). ADCP activity was not found for various doses of 651-LALA opsonized with Cal/09 (Fig 4G and 4H). Therefore, we demonstrated that 651 possesses ADCC and ADCP activity *in vitro*.

It has been shown that broadly neutralizing antibodies against the stem region, but not globular head domain, of HA can elicit neutrophil-mediated ADCP [37]. We further examined whether 651 induces ADCP by neutrophils. We found that 651 was unable to mediate the uptake of Cal/09 virus by human peripheral neutrophils (S3C–S3E Fig), in line with the

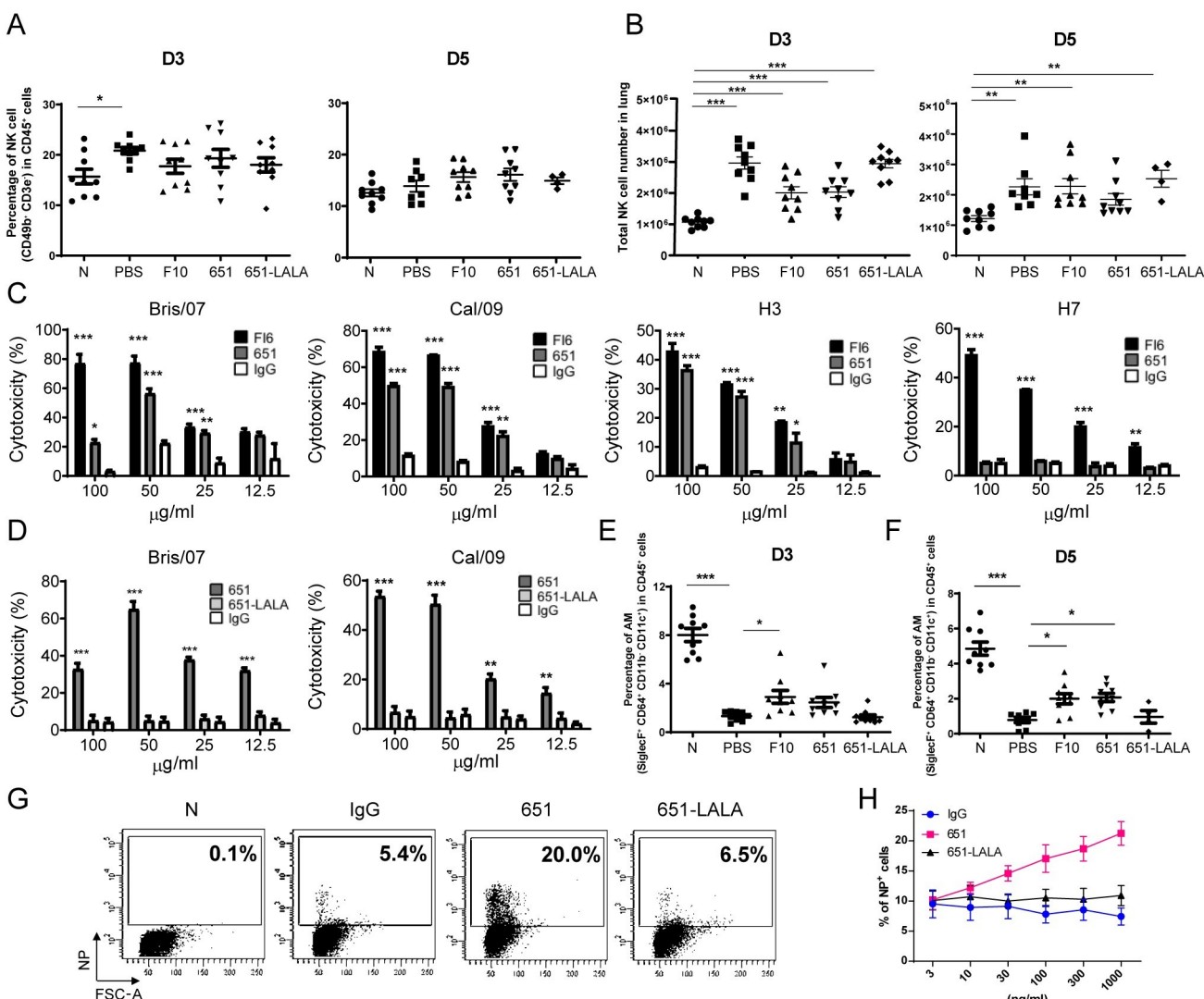

**Fig 4. 651 possessed ADCC or ADCP against influenza viruses *in vitro*.** (A and B) FACS analysis showing the percentage (A) and number (B) of pulmonary NK cells (CD49b$^+$) in CD45$^+$CD3e$^-$-gated populations in indicated mAb-pretreated mice on day 3 and day 5 post challenge of Cal/09 virus. n = 4–9 mice per group in A and B. (C) 651 showed ADCC against H1 (Bris/07 and Cal/09) and H3 but not H7-expressing 293T cells. (D) 651-LALA mutation abolished ADCC mediated by 651 to kill cells expressing H1 from Bris/07 or Cal/09 viruses. IgG was used as the negative control in C and D. n = 3 in C and D. (E and F) FACS analysis showing the percentage of alveolar macrophages (AMs) (SiglecF$^+$CD64$^+$) in CD45$^+$CD11b$^-$CD11c$^+$ gate in indicated mAb- or PBS-pretreated mice on day 3 (E) and day 5 (F) post-challenge with Cal/09 virus. n = 9 mice per group in E and n = 4–9 mice in F. (G) FACS showing the NP positive signals in sialidase (0.5 unit/mL) treated THP-1 cells incubated with 651-, 651-LALA-, or IgG- (all at 100 ng/mL) opsonized Cal/09 virus. Percentage of NP positive cells is indicated. (H) FACS results showing the percentage of NP positive signals in sialidase treated THP-1 cells incubated with serial doses of 651 or 651-LALA mAbs and Cal/09 virus. n = 4 in IgG and 651-LALA groups and n = 5 in 651 group. N represents naïve mice without infection. Results are mean ± SEM. $^*p < 0.05$; $^{**}p < 0.01$; $^{***}p < 0.001$.

previous finding that immune complexes formed by HA head-specific antibodies did not induce ADCP by neutrophils [37].

## NK cells and alveolar macrophages coordinate the antiviral responses mediated by 651

Having demonstrated the NK cell-mediated ADCC and macrophage-mediated ADCP mediated by 651 *in vitro*, we next examined the involvement of NK cells and alveolar macrophages

in 651-mediated protection during virus challenge *in vivo*. Mice were administered an anti-ASGM1 antibody to deplete NK cells (S4A Fig) and intranasally administered clodronate liposomes to deplete alveolar macrophages (S4A Fig). Intranasal administration of control liposome reduced some protective effects of F10 and 651 through unidentified mechanisms, likely because liposomes may block phagocytosis or affect macrophage functions to some extent [38]. Therefore, administration of the control liposome may not represent the best control for clodronate liposome treatment. Accordingly, our results support the notion that, as compared with the sham treatment, control liposome treatment substantially reduced the frequency of alveolar macrophages (S4A Fig). Therefore, unlike F10, which is a potent neutralizing mAb, the protective efficacy of 651 was affected in control liposome + control antibody treated mice because the activity of 651 depends on ADCC (Fig 5A, third panel). Nevertheless, both F10 and 651 still possessed prophylactic activity with statistical significance in Cal/09-infected mice injected with the control antibody or liposomes, as shown by the significantly improved survival (Fig 5A). Remarkably, the prophylactic effects of 651 diminished when NK cells and alveolar macrophages were co-depleted (Fig 5B), while 651 still protected mice lacking either NK cells or alveolar macrophages. In contrast, F10-administered mice remained resistant to influenza virus challenge, even when NK cells and alveolar macrophages were co-depleted (Fig 5B), implying the predominant neutralizing activity of F10 [25]. The body weight loss results demonstrated that the co-depletion of NK cells and alveolar macrophages was required to eradicate the protective effect of 651 (S4B and S4C Fig). Consistently, the production of IFN-β and proinflammatory cytokines, such as IL-6 and MCP-1, in the lungs was comparable between the 651- and PBS-pretreated mice co-depleted with NK cells and alveolar macrophages on day 3 (Fig 5C) and day 5 (Fig 5D) post infection. The viral titers on day 5 post infection were also similar between the PBS- and 651-pretreated mice co-depleted with NK cells and alveolar macrophages (Fig 5E). These results suggest that the protective efficacy of 651 *in vivo* could be attributed to both NK cells and alveolar macrophages.

Although Fc-FcγR interactions utilized by non-neutralizing anti-HA head mAbs to mediate protection *in vivo* have been demonstrated [39], which types of immune cells confer the protection remained to be defined. From our study, we concluded that a non-neutralizing mAb, which has a broad spectrum of recognition range and is able to employ NK cells and alveolar macrophages for effector functions, alleviates the inflammatory responses and reduces the mortality associated with influenza viral infection. This broad-spectrum mAb was isolated from HA$_{mg}$-immunized mice. It remains to be ascertained if a natural influenza viral infection or influenza vaccination in humans can elicit a significant abundance of such broad-recognition and non-neutralizing anti-HA antibodies that confer protection through effector functions. One recent study demonstrated the antibody-dependent enhancement of influenza virus infection involved the recognition of the head domain and the promotion of viral fusion [40]. Studying the structure of HA, 651, and the FcR tertiary complex will further reveal the allosteric preferences of engaging target cells and effector cells for effective ADCC. It has been shown that HA-specific antibodies require a second intermolecular interaction to induce optimal effector functions, called "two points of contact" which involves the interaction between receptor-binding domain of HA and sialic acids on the effector cells in addition to the engagement of Fc of antibody and FcR on the effector cells [41]. Given that 651 binds with the globular head region, the ability of 651 to mediate Fc-dependent effector functions is also likely due to promoting the interaction between HA and sialic acid receptors on the effector cells. Nevertheless, the Fab of 651 recognizes the globular head region, near the glycan modification sites of HA, whose sequences are conserved in Bris/07 and Cal/09. We consider that removal or truncation of the glycan may allow the exposure of hidden epitopes of HA that are relatively conserved across influenza viruses.

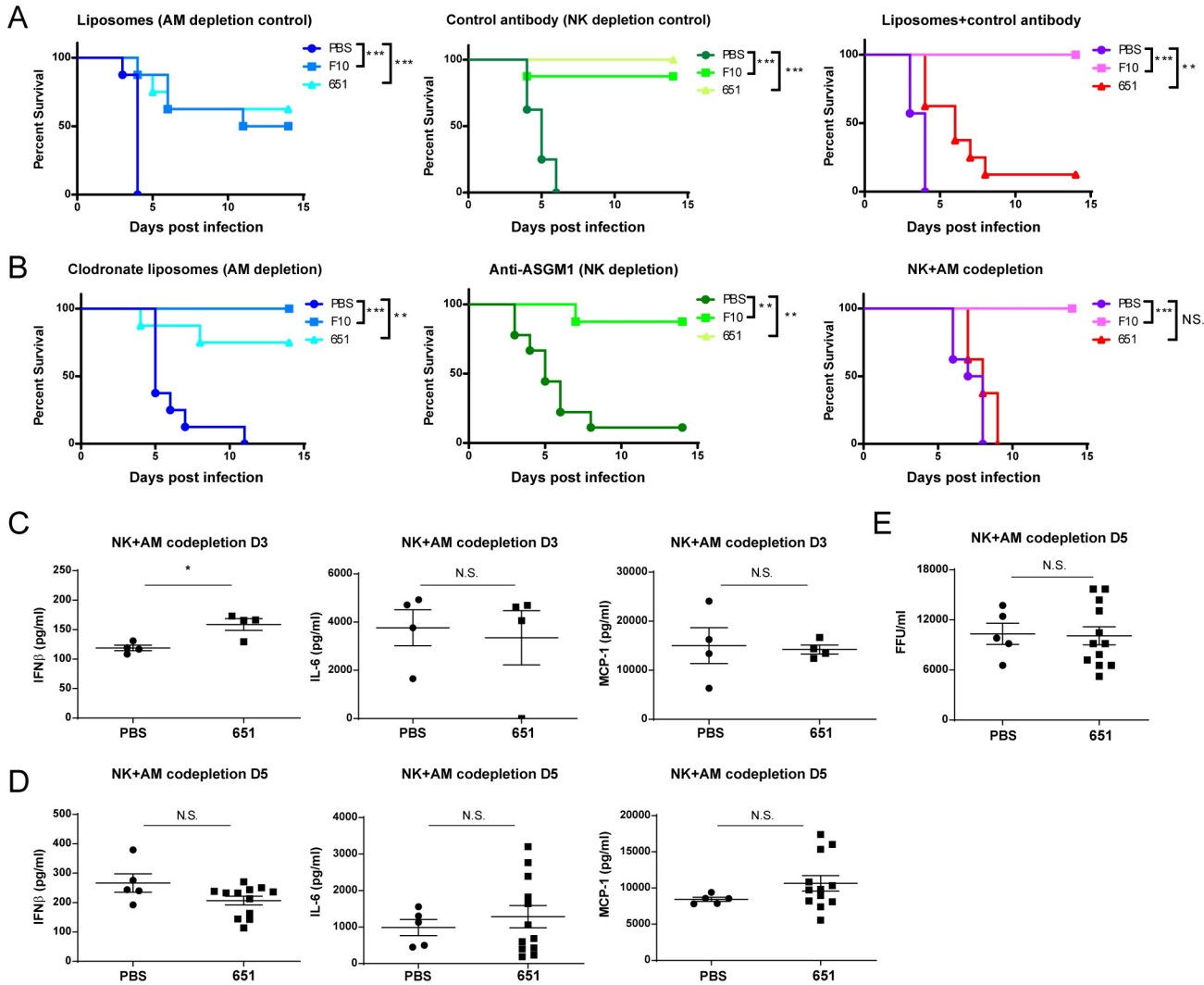

**Fig 5. 651-mediated protection in mice requires both alveolar macrophages and NK cells.** (A and B) Cumulative survival rate of mice receiving control reagents, including liposome alone, control antibody, or both (A), or clodronate liposomes or/and anti-ASGM1 antibody (B) prior to i.p. injection of mAbs and 100 LD50 Cal/09 intranasal challenge. n = 7–8 mice per group in A and 8–9 mice per group in B. (C and D) Comparison of pulmonary IFNβ, MCP-1, and IL-6 protein levels measured by ELISA in alveolar macrophages (AM)-NK co-depleted mice treated with either PBS or 651 on day 3 (C) and day 5 (D) post-Cal/09 infection. n = 4 mice per group in C. n = 5 in PBS group and n = 12 in 651 group in D. (E) Fluorescent focus assay showing the comparable influenza virus titers in AM-NK co-depleted mice treated with either PBS or 651 on day 5 after Cal/09 infection. n = 5 in PBS group and n = 12 in 651 group in E. Data in A and B were analyzed by log-rank (Mantel-Cox) test. Data in C-E are mean ± SEM. *p < 0.05; **p < 0.01; ***p < 0.001. N.S. = no significant difference.

Accumulating efforts have improved the potency of antibody-mediated effector functions by generating Fc variants or modulating the glycan compositions of Fc [42–44]. Further modifications of 651 may optimize its protection efficacy *in vivo*. We found that, in addition to NK-cell-mediated ADCC, alveolar macrophage-mediated ADCP is also critical to the protective effects of 651. In response to viral infections, alveolar macrophages are activated, becoming phagocytic and producing large quantities of inflammatory cytokines, including type I IFNs [45]. Some mAbs have been demonstrated to confer protection against influenza viral infection through the action of antibody-induced inflammation and ADCP mediated by alveolar macrophages [19]. However, neither F10 nor 651 increased the production of inflammatory cytokines in the lungs during influenza virus infection, possibly because of the differential

cytokine kinetics resulting from the different doses and types of virus used. Although awaiting confirmation in future studies, it is also plausible that the antibody affinity in this study selectively modulated certain subsets of the heterogeneous alveolar macrophages, which are, at minimum, composed of resident macrophages that populate the lungs during embryogenesis and circulating blood monocytes recruited to the lungs following infection [46].

In conclusion, the study described herein revealed the protective effects and the underlying mechanisms of a broad-recognition and non-neutralizing mAb against influenza infection, which may prove valuable in future assessments of vaccine efficacy and therapeutic antibody development.

## Materials and methods

### Ethics statement

All animal experiments were evaluated and approved by the Institutional Animal Care and Use Committee of Academia Sinica (Animal Protocol 14-03-656).

### Influenza viruses, HA expression plasmids and HA proteins

The vaccine strains of influenza viruses A/H1N1/California/07/2009 (Cal/09), A/H1N1/Brisbane/59/2007 (Bris/07), A/H5N1/Vietnam/1194/2004/NIBRG14 (NIBRG14) and A/H3N2/Victoria/361/2011 (Vic/11) were from the reference collection of the National Institute for Biological Standards and Control. All viruses were inoculated into the allantoic cavities of 10-d-old specific pathogen-free embryonated chicken eggs for 2 days at 35˚C. The 50% tissue culture infective dose ($TCID_{50}$) of viruses in Madin-Darby canine kidney (MDCK) (CCL-34; American Type Culture Collection) cells and $LD_{50}$ of virus in BALB/c mice were determined before experiments. Methods for expression and purification of the ectodomains of HA from Cal/09, Bris/07, NIBRG14, Vic/11, A/H1N1/WSN/1933 (WSN), A/H1N1/Puerto Rico/8/1934 (PR8), A/H7N9/Shanghai/2/2013, and B/Victoria/2/87 were described previously [23]. Methods for generation and purification of stem region of HA from Bris/07 as well as $HA_{mg}$ and $HA_{fg}$ from Cal/09 and Bris/07 were described previously [24,47].

### Flow cytometry

To determine the immune cell subsets in lungs of infected mice, whole lung tissues were removed, followed by isolation of single cell suspensions using the lung dissociation kit (Miltenyi Biotec). Cells were harvested and suspended in FACS buffer (2% FBS in PBS) at a density of $10^{6}$/ml. The antibodies used in this study are anti-mouse B220 antibody (BD, clone RA3-6B2), anti-mouse IgG1 antibody (BD, clone X56), anti-mouse CD38 antibody (Biolegend, clone 90), anti-mouse CD45 antibody (BD, clone 30-F11), anti-mouse CD49b antibody (Biolegend, clone DX5), anti-mouse CD3e antibody (BD, clone 145-2c11), anti-mouse Siglec-F antibody (BD, clone E50-2440), anti-mouse CD64 antibody (BD, clone X54-5/7.1), anti-mouse CD11b antibody (BD, clone M1/70), anti-mouse CD11c antibody (Biolegend, clone N418), anti-human CD11b (BD, clone ICRF44) and anti-human CD66b (BD, clone G10F5). Cellular fluorescence intensity was analyzed by FACSCanto (BD Biosciences) and FCS Express 3.0 software.

### Cloning and expression of HA-specific mAbs

Ig genes from a single B cell were isolated primarily following the protocols reported previously [24,48,49]. Briefly, single cell was collected from the spleens of BALB/c mice immunized i.m. two times at 2 weeks apart with 20 μg Bris/07 $HA_{fg}$ or $HA_{mg}$. The

$HA_{fg}^+B220^+IgG1^+CD38^+$ splenic B cells were isolated on day 29 by a cell sorter (FACSAria II), and the single B cell was sorted into 96-well PCR plates (Thermal Scientific). RT-PCR reactions were performed as described previously [24,48,49]. Aliquots of nested PCR products were sequenced and analyzed using IMGT/V-Quest (http://www.imgt.org) to identify the highest homology gene loci of germ-line V, D, and J genes. Those candidate Ig heavy- and light-chain cDNA segments were further subcloned to a chimeric Ig expression vector modified from the tandem chimeric antibody expression (TCAE) vector and the pIgG1(κ) vector (provided by Dr. T. W. Chang, Genomics Research Center, Academia Sinica, Taipei, Taiwan). The generation of the mutations at Fc region of mAb 651, Leu234Ala/Leu235Ala (LALA), was performed by substituting two leucine (L) residues with alanine (A) at a.a. 234 and 235 through site-directed mutagenesis of cDNA. Vector for mAb C05 expression [7] was kindly provided by Dr. T. W. Chang. The positive control of anti-HA antibodies FI6 [6] and F10 [25] were expressed and purified following previously described procedures [50].

## Purification and binding efficacy of recombinant mAbs

Ig expression vector was transfected into HEK293F cells by using Expi293 Expression System Kit (Thermo Fisher Scientific). Three days later, supernatant was collected for antibody purification by using Protein A Sepharose (GE) chromatography. Antibody was dissolved in PBS. Enzyme-linked immunosorbent assay (ELISA) was used to determine the binding of recombinant mAbs with HA. Briefly, purified HA of Cal/09, Bris/07, WSN/33, PR8/34, H3, H5, H7 and Flu B viruses was coated on the 96-well plates (0.1 µg/well in 100 µL) for 2 h. The HA-coated plates were then incubated with the 2-fold serial dilutions of recombinant mAbs starting from the highest concentration at 0.05 µg/mL for 2 h. The captured recombinant antibodies were detected by HRP-conjugated anti-human antibodies and peroxidase substrate solution substrate (BD Biosciences). Competitive ELISA was performed with biotin-labeled 651 (labeled with One-Step Antibody Biotinylation Kit (Miltenyi Biotec)). Biotinylated 651 (200 ng/mL) was added into HA-precoated ELISA plates with serially diluted competitor F10 or C05 mAb, or HA protein, and plates were then incubated at 37˚C for 1 h. The plates were washed and then further incubated with 100 µl of 50-fold diluted Avidin-HRP (R&D Systems) for another 1 h at 37˚C. The plates were developed with TMB method and OD was read at 450 nm by the SpectraMax M2 Microplate Reader (Molecular Devices). The bio-layer interferometry (BLI) was performed on Octet RED 96 instrument (FortéBio, Inc.). Antibody was immobilized onto anti-human AHC biosensors (FortéBio, Inc.) and incubated with HAs at 0.013–3.17 µM for 90 seconds for association and then incubated in 20 mM Tris, pH 8.0, 150 mM NaCl, 0.005% Tween 20 for dissociation for 90 seconds. The signals for each binding event were measured with a 1:1 Langmuir binding model for $k_{on}$, $k_{off}$ and $K_d$ value determination.

## Microneutralization assay

Microneutralization assay was performed as previously described [44]. The freshly prepared virus was quantified with the median TCID50. The 10-fold TCID50 of virus was mixed in equal volume with 2-fold serial dilutions of mAbs in 96-well plates and incubated for 1 h at 37˚C. The mixture was added onto the MDCK cells ($1.5 \times 10^4$ cells per well) in the plates followed by incubation at 37˚C for 16–20 h. The cells were washed with PBS, fixed in acetone/ methanol solution (1:1 vol/vol), and blocked with 5% (wt/vol) skim milk in PBS. Quantification of virus was detected by ELISA with a polyclonal antibody against influenza A nucleoprotein (NP) protein. The anti-NP primary antibody was added and incubated for 1 h at 37˚C. After washing with PBST (PBS + 0.01% Tween 20), the secondary antibody (rabbit anti-goat IgG HRP conjugated) was added and incubated for 1 h at 37˚C. Peroxidase substrate solution

was then added and incubated for 15 min at room temperature in the dark, followed by adding stop solution. The absorbance (OD) of the wells was read at 450/620 nm.

## Hemagglutination inhibition assay (HI) assay

The hemagglutination units of Cal/09 virus were determined following a previous report [24]. Briefly, the 2-fold serial dilutions of Cal/09 virus were added into the 96-well round (U) bottom plates, and 0.2% turkey red blood cells (Jianrong Farm, Taiwan) were added into each well and mixed well. After 30 min, the hemagglutination unit (HAU) was identified. For antibody mediated hemagglutination inhibition detection, 1024 HAU of Cal/09 virus was mixed with 2-fold serial dilutions of mAbs and incubated for 30 min. After incubation, 0.2% turkey red blood cells were added into each well and mixed well. After 30 min, the hemagglutination inhibition was determined.

## Virus challenge in mice

Female BALB/c mice, purchased from National Laboratory Animal Center, Taiwan, at 8 wk old were injected i.p. with 300 μg (15 mg/kg) or 150 μg (7.5 mg/kg) purified mAb 2 h before or 24 h after intranasal challenge with H1N1 Bris/07 or Cal/09 virus with a lethal dose (100 LD50). In some experiments, FcγR knockout mice (purchased from The Jackson Laboratory) at 8 wk old were used. Mouse body weight and survival data were measured and recorded every day afterwards. In some experiments, NK cells were depleted by anti-ASGM1 (20 μL/mouse, WAKO) antibody or PBS (as the control) by ip injection 2 d before mAb injection and 1 and 4 d post-infection. Macrophages were depleted by clodronate liposomes (100 μL/mouse, ClodronateLiposomes) 2 and 4 d before mAb injection and 1 d post-infection.

## ADCC assay

ADCC was performed as previously reported [44]. HEK293T cells transfected with HA-expression vectors (Bris/07, Cal/09, H3, and H7) for 48 h and human peripheral blood mononuclear cells (PBMCs) were used as target cells and effector cells, respectively. Human PBMCs from healthy donors were obtained from Taipei Blood Center with the consent procedures approved by the Academia Sinica Research Ethics Committee. PBMCs were isolated by density gradient centrifugation with Ficoll-Paque at 400×$g$ for 30 min without brake at 22˚C. Control IgG was purchased from GeneTex (GTX16193). mAbs at 2-fold serial dilutions were added to the co-culture composed of $5\times10^3$ 293T cells transiently expressing HA proteins from indicated strains of influenza viruses and $2.5\times10^5$ effector PBMCs (E:T ratio = 50), and incubated for 5 h at 37˚C. The supernatant of the co-culture was collected and analyzed by CytoTox 96 Non-Radioactive Cytotoxicity Assay Kit (Promega).

## ADCP assay

Isolation of human neutrophils from peripheral blood was carried out by using Ficoll-Paque PLUS density gradient (GE Healthcare) according to the procedures described previously [51]. ADCP by THP1 cells (from ATCC) and neutrophils was performed essentially as a previous report [52]. Control IgG was purchased from GeneTex (GTX16193). Briefly, Cal/09 virus was incubated with indicated antibodies at 37˚C for 1 h and added to the sialidase (0.5 unit/mL, Sigma) pre-treated THP-1 cells or neutrophils. After 1 h incubation, THP-1 cells or neutrophils were washed three times with RPMI and incubated at 37˚C. After 6 h incubation, THP-1 cells or neutrophils were fixed in 4% paraformaldehyde, permeabilized and stained with anti-

influenza NP antibody (Abcam, Ab20921, 1:100 dilution). The levels of phagocytosis were monitored by flow cytometry.

## Histology analysis, cytokine levels and viral titers in lung analysis

The perfused mouse lungs isolated at day 3 and day 5 after infection were fixed in 4% formaldehyde, embedded in paraffin and cut into 3 μm-thick sections for Hematoxylin & Eosin (H&E) staining. To determine the viral titers in the infected lungs, infected mice were sacrificed on day 3 and day 5 post-infection. The whole lungs were collected and homogenized in PBS (2.5 mL/g lung). The viral titers from the homogenates were determined by immunoplaque assay on MDCK cells following previously established protocols [53]. Briefly, MDCK cells were seeded into 24-well plates. One day later, the MDCK monolayers were washed once with PBS. Ten-fold serial dilutions of lung homogenates, made in in serum-free RPMI containing 0.5 μg/mL TPCK-trypsin, were added to the MDCK cells. After 1 h incubation, the cells were washed once with PBS and overlaid with DMEM containing 0.5 μg/mL TPCK-trypsin and 0.5% agarose. After overnight incubation, MDCK cells were fixed with 10% formaldehyde and the agarose were removed. The cells were fixed, permeabilized and stained with anti-influenza NP antibody and the focus forming units were determined. The lung homogenates were used for determining the levels of cytokines by using IFN-β and IFN-α ELISA kits from PBL Assay Science, MCP-1 and TNF-α ELISA kits from eBioscience, IL-6 ELISA Kit from ABclonal Inc., and IFN-λ2/3 ELISA kit from R&D Systems.

## Hydrogen-deuterium exchange-mass spectrometry (HDX-MS) assay

HDX-MS was performed essentially as previously reported [54]. Briefly, the Fab regions of the antibodies were prepared by using Pierce Fab Preparation Kit (Thermo). HA protein (from Bris/07) and Fab region of antibody were co-incubated for 30 min at 37˚C, digested by PNGase F (NEB) for 2 h at 37˚C. Hydrogen was exchanged to deuterium by adding $D_2O$ to the reaction for 10 and 20 min at room temperature and stopped the exchange by increasing urea concentration to 2 M and low temperature (4˚C). After pepsin and protease type XIII digestion, the sample was ready for MALDI TOF analysis with ESI mass spectrometry (Velos Pro LTQ, Thermo Scientific, mass spectrometric core facility, Genomics Research Center, Academia Sinica) to determine the interacting region between HA and Fab.

## Statistical analysis

Data are shown as the mean ± SEM. Statistical analyses were performed using GraphPad Prism 8 software. Analysis of differences between two groups was performed by an unpaired two-tailed Student's t test. Comparisons between multiple groups were performed using a one-way ANOVA, followed by Dunnetts' honestly significant difference post hoc test. Survival rate differences between groups were analyzed by using log-rank (Mantel-Cox) test.

## Supporting information

**S1 Fig.** *In vitro* **characterization of mAb 651-binding to HA.** (A) HDX-MS analysis map of the Bris/07 HA-binding sites of mAb 651 (yellow). N-glycosylation sites are marked in red. (B) Multiple sequence alignment of HA amino acid sequences of indicated influenza virus strains using CLUSTALW (website: https://www.genome.jp/tools-bin/clustalw). mAb-bound amino acids mapped by HDX-MS are highlighted in yellow, and N-glycosylation sites on Bris/07 and Cal/09 are labeled in red. (C) The binding epitopes of mAb F10 (left) and mAb C05 (right) on HA are labeled in red. (D) Competitive ELISA showing various concentrations of mAb F10 or

mAb C05 did not compete the binding of 651 with Cal/09 HA (left panel) or Bris/07 HA (right panel). Effective competition of 651 binding by full length Cal/09 (left panel) or Bris/07 HA (right panel) at two different doses was also indicated. Results are mean ± SEM (n = 3 for the F10 and C05 groups). (E) mAb 651 did not provide hemagglutination inhibition (HI) activity. The HI titers of Cal/09 virus were determined by incubation of 0.2% turkey red blood cells with indicated mAb at 2-fold serial dilutions. F10 and IgG served as the positive control and negative control for HI activity, respectively.
(TIF)

**S2 Fig. Weight changes of mice pretreated or post-treated with mAbs, followed by influenza virus infection.** (A and B) Mice were pretreated (i.p.) with indicated mAbs 2 h before intranasal challenge with 100 LD50 of H1N1 Bri/07 virus (A) or Cal/09 virus (B). Body weights were monitored daily for 14 days. n = 5–8 mice per group. (C and D) Mice were i.p. pretreated with indicated mAbs 24 h after intranasal challenge with 100 LD50 of H1N1 Bri/07 virus (C) or Cal/09 virus (D). Body weights were monitored daily for 14 days. n = 4–8 mice per group. (E) Unlike 651, 651-LALA-pretreatment did not prevent weight loss caused by Cal/09 infection. n = 8 mice per group. (F) Weight changes in FcγR knockout (KO) and control WT mice were observed after F10 or 651 pretreatment and Cal/09 infection. F10 mAb was used as the positive control. n = 6–8 mice per group.
(TIF)

**S3 Fig. Neutrophils did not involve in mAb 651-mediated ADCP *in vitro*.** (A) FACS showing the NP positive signals in sialidase (0.5 unit/mL) treated THP-1 cells incubated with 651, F10, or IgG (all at 3,000 ng/mL) opsonized Cal/09 virus. Percentage of NP positive cells is indicated. (B) FACS results showing the percentage of NP positive signals in sialidase treated THP-1 cells incubated with serial doses of 651, F10 or IgG opsonized Cal/09 virus. (C) FACS showing the purity of neutrophils (CD11b$^+$CD66b$^+$) isolated from human peripheral blood by using dextran sedimentation and Ficoll-Hypaque density gradient centrifugation. (D) FACS showing the NP positive signals in sialidase (0.5 unit/mL) treated THP-1 cells (upper panel) and neutrophils from human peripheral blood (lower panel) incubated with 651-, 651-LALA-, or IgG- (all at 1,000 ng/mL) opsonized Cal/09 virus. Percentage of NP positive cells is indicated. (E) FACS results showing the percentage of NP positive signals in sialidase treated THP-1 cells and human neutrophils (Neu) incubated with serial doses of 651, 651-LALA mAbs or IgG opsonized Cal/09 virus. Results are mean ± SEM (n = 3 in B and E).
(TIF)

**S4 Fig. Depletion efficiency of NK cells and alveolar macrophages (AMs), and the weight changes of mice depleted with AMs and/or NK cells after mAb pretreatment and Cal/09 virus challenge.** (A) Flow cytometric analysis showing a significant reduction in the frequency of alveolar macrophages (AMs, SiglecF$^+$CD11c$^+$CD11b$^-$ CD64$^+$) or NK cells (CD49b$^+$CD3e$^-$) in lungs on day 5 after employing the control or depletion reagents and infection. (B and C) Weight changes of mice treated with control reagents (B) or depleted with AMs and NK cells (C) after pretreatment of 651 (300 μg/dose), F10 (300 μg/dose), or PBS for 2 h before intranasal challenge with 100 LD50 of H1N1 Cal/09 virus. Results are mean ± SEM. n = 8–9 mice per group in B and C.
(TIF)

## Acknowledgments

We thank Shii-Yi Yang and Szu-Teng Ma for excellent technical support.

## Author Contributions

**Conceptualization:** Kuo-I Lin.

**Data curation:** Yi-An Ko, Yueh-Hsiang Yu, Yen-Fei Wu, Yung-Chieh Tseng, Chia-Lin Chen, King-Siang Goh, Hsin-Yu Liao, Ting-Hua Chen.

**Formal analysis:** Yi-An Ko, Yueh-Hsiang Yu, Yen-Fei Wu, Yung-Chieh Tseng, Chia-Lin Chen, King-Siang Goh.

**Funding acquisition:** Chi-Huey Wong, Kuo-I Lin.

**Methodology:** Yi-An Ko, Yueh-Hsiang Yu, Ting-Jen Rachel Cheng, An-Suei Yang, Che Ma.

**Resources:** Ting-Jen Rachel Cheng, An-Suei Yang, Chi-Huey Wong, Che Ma.

**Supervision:** Kuo-I Lin.

**Writing – original draft:** Yi-An Ko, Yueh-Hsiang Yu, Kuo-I Lin.

**Writing – review & editing:** Kuo-I Lin.

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
