## [Decision Letter · Decision Letter 0]

5 Jan 2021

Dear Dr. Lin,

Thank you very much for submitting your manuscript "A non-neutralizing antibody broadly protects against influenza virus infection by engaging effector cells" for consideration at PLOS Pathogens. As with all papers reviewed by the journal, your manuscript was reviewed by members of the editorial board and by several independent reviewers. In light of the reviews (below this email), we would like to invite the resubmission of a significantly-revised version that takes into account the reviewers' comments.

Please pay particular attention to the suggestions from Reviewers 2 and 3, respectively, pertaining to the role of neutrophils in ADCP and defining the HA epitope that binds 651.

We cannot make any decision about publication until we have seen the revised manuscript and your response to the reviewers' comments. Your revised manuscript is also likely to be sent to reviewers for further evaluation.

Sincerely,

Sabra L. Klein

Associate Editor

PLOS Pathogens

Carolina Lopez

Section Editor

PLOS Pathogens

Kasturi Haldar

Editor-in-Chief

PLOS Pathogens

orcid.org/0000-0001-5065-158X

Michael Malim

Editor-in-Chief

PLOS Pathogens

orcid.org/0000-0002-7699-2064

Please pay particular attention to the suggestions from Reviewers 2 and 3, respectively, pertaining to the role of neutrophils in ADCP and defining the HA epitope that binds 651.

Reviewer's Responses to Questions

**Part I - Summary**

Reviewer #1: In general, data presented in this manuscript appear solid and support the conclusion that non-neutralizing antibody is able to have a protective activity against influenza infection via ADCC and ADCP. The authors generated a non-neutralizing antibody, 651, from influenza-infected mice and showed that 651 has little neutralizing activity using in vitro assay. Subsequently, the authors demonstrated that 651 treatment promotes survival of influenza-infected mice and reduces lung inflammation. The authors examined the protective activity of 651 is via ADCC and ADCP, which appear to be dependent on alveolar macrophages and NK cells. Overall, I think the manuscript is acceptable for publication once some corrections and clarifications are addressed.

Reviewer #2: The manuscript by Ko et al. describes a non-neutralizing antibody (651) that binds to the HA head domain and mediates protection in vivo via the induction of Fc-dependent effector function of alveolar macrophages and NK cells. In general, the study is very well conducted and adds to a growing body of literature demonstrating the importance of non-neutralizing antibodies in protection against influenza virus infection. The only notable weakness is the lack of attention given to neutrophils, a potentially important and abundant Fc-bearing cell type. Other minor comments for the authors’ consideration are noted below.

Reviewer #3: This manuscript by Ko et al. describes a broadly cross-reactive 651 mAb that binds to an undefined epitope in the globular head domain of HA and can recognize an array of group 1 & 2 influenza viruses. Using a combination of in vitro assays and mouse work, the authors demonstrate that 651 does not protect mice through virus neutralization but rather through ADCC and ADCP. The novelty of this study is questionable given that this has already been widely reported by many other studies. There are, however, some novel aspects of this study like the NK cell and macrophage co-depletion assays. A major weakness of this study is that there was no attempt made by the authors to define the HA head epitope bound by 651. General execution of this study was sufficient overall, but there were some unexplained results from the co-depletion and cytotoxicity assays that are not satisfactorily addressed.

**Part II – Major Issues: Key Experiments Required for Acceptance**

Reviewer #1: None.

Reviewer #2: 1. Neutrophils are a highly abundant FcR-expressing leukocyte. They have also been shown to mediate ADCP against influenza virus in vitro (PMID: 27703076) but the importance of their contribution to antibody-mediated protection in vivo is less clear. The authors glaze over the possible contributions of these cells in their manuscript. It would be interesting to perform in vitro ADCP assays using 651 and neutrophils to determine whether they are capable of mediating ADCP. It would also be interesting for the authors to perform a neutrophil depletion experiment in the context of 651 passive transfer (as they did for NK cells and alveolar macrophages) to test the possible contribution of these cells to protection in their system.

Reviewer #3: Major concerns:

1. Studies that describe a single novel mAb typically define the exact HA epitope recognized using structural work or at a minimum a competition ELISA. In order for this manuscript to be considered for publication, the exact HA epitope bound by 651 should be established.

2. Why is the 651 mAb not directly compared to an anti-HA stem bNAb (like CR9114 for example) if the authors suspect Fc-mediated effector functions may be important for protection in the mouse model? It has been repeatedly demonstrated that Fc-mediated functions, like ADCC and ADCP, are required for protection by anti-HA stem bNAbs in vivo. Therefore, an anti-HA stem bNAb should be included as a positive control across all the assays performed instead of OR in addition to F10.

3. Previous studies have described broadly binding, non-neutralizing mAbs targeting the globular HA head that are protective in murine models of influenza (DiLillo et al 2016), so this aspect of the study is not particularly novel.

4. The most novel aspect of this study, in my opinion, is the NK cell and alveolar macrophage co-depletion data. These co-depletion assays should be performed with other broadly binding HA mAbs (HA stem mAbs and other globular head mAbs with known epitopes) to determine whether NK cells and alveolar macrophages are involved in mediating protection against other HA epitopes OR if this is unique to the epitope recognized by the 651 mAb.

5. Why is there a massive decrease in survival for the control liposome + control antibody mice that received 651 (Figure 5A, 3rd panel)? In this group, only 10% of mice that got 651 survived compared to 100% of mice that received F10. Why would the control liposomes + control antibody have this effect on 651 treated mice? This finding is completely ignored in the Results/Discussion and needs to be addressed. It is very strange the control liposomes and control antibody would have this kind of impact on protection by 651 and suggests some kind of experimental issue (possibly with the liposomes).

6. What is this control IgG that was used as the negative control in the ADCC and ADCP assays? Some details about this are required to know if this is a suitable negative control, especially since some background killing was observed in ADCC assay (20% in 50ug/ml wells for Bris/07). What was the % cytotoxicity in the no antibody control wells of the cytotoxicity assay?

**Part III – Minor Issues: Editorial and Data Presentation Modifications**

Reviewer #1: See the attachment.

Reviewer #2: 1. Line 44 (and elsewhere): “flu” is too colloquial for scientific publication – please edit to “influenza”

2. Line 45: Influenza A viruses are now known to encode up to at least 14 proteins (not 11) – please correct

3. Line 47: There are 4 subtypes of influenza virus: A,B,C,D (not 3) – please correct

4. Line 60: The lower neutralization capacity of HA stem-binding bnAbs is directly shown in PMID: 25589655

5. Lines 69-71: ADCP mediated by alveolar macrophages has also been shown to have importance in PMID: 29018261

6. Line 72: The authors cite papers showing the impact of reducing glycosylation on antigenicity – it might be worth noting that hyperglycosylation can also influence antigenicity as shown in PMID: 24155380

7. Lines 183-184: The authors should note that antibodies against NA have also been reported to elicit ADCC, though less potently than those that bind the HA stem (PMID: 27698132).

8. The ability of 651 to mediate Fc-dependent effector functions is likely due to its ability to preserve the “two points of contact” previously shown to be essential for HA-specific antibody-mediated ADCC (PMIDs: 27698132, 27647907). This mechanistic explanation would be worth mentioning in the discussion.

Reviewer #3: Minor issues:

1. Line 47: Influenza D also exists.

2. Lines 49-50: Saying that seasonal influenza vaccination protects 2/3rds of people vaccinated is not accurate. This varies tremendously year-to-year depending on how well the vaccine strains of influenza virus match the strains of influenza virus circulating in the population.

PLOS authors have the option to publish the peer review history of their article (what does this mean?). If published, this will include your full peer review and any attached files.

Reviewer #1: No

Reviewer #2: No

Reviewer #3: No
---

## [Decision Letter · Decision Letter 1]

2 May 2021

Dear Dr. Lin,

Thank you very much for submitting your manuscript "A non-neutralizing antibody broadly protects against influenza virus infection by engaging effector cells" for consideration at PLOS Pathogens. As with all papers reviewed by the journal, your manuscript was reviewed by members of the editorial board and by several independent reviewers. The reviewers appreciated the attention to an important topic. Based on the reviews, we are likely to accept this manuscript for publication, providing that you modify the manuscript according to the review recommendations.

Your revision does not address reasonable concerns raised by Reviewer 3. Please address the experimental concerns below in the Results as well as in the interpretation of these data in the Discussion.

Sincerely,

Sabra L. Klein

Associate Editor

PLOS Pathogens

Carolina Lopez

Section Editor

PLOS Pathogens

Kasturi Haldar

Editor-in-Chief

PLOS Pathogens

orcid.org/0000-0001-5065-158X

Michael Malim

Editor-in-Chief

PLOS Pathogens

orcid.org/0000-0002-7699-2064

Your revision does not address reasonable concerns raised by Reviewer 3. Please address the experimental concerns below in the Results as well as in the interpretation of these data in the Discussion.

Reviewer Comments (if any, and for reference):

Reviewer's Responses to Questions

**Part I - Summary**

Reviewer #1: The author's revision appears satisfactory.

Reviewer #2: The authors have addressed all concerns raised in the initial round of reviews. This is a very solid study that adds to a growing body of literature supporting a role of Fc-dependent effector functions in the protection mediated by broadly-reactive antibodies against HA.

Reviewer #3: This manuscript by Ko et al. describes a broadly cross-reactive 651 mAb that binds to an undefined epitope in the globular head domain of HA and can recognize an array of group 1 & 2 influenza viruses. Using a combination of in vitro assays and mouse work, the authors demonstrate that 651 does not protect mice through virus neutralization but rather through ADCC and ADCP. The novelty of this study is questionable given that this has already been widely reported by many other studies. There are, however, some novel aspects of this study like the NK cell and macrophage co- depletion assays. A major weakness of this study is that there was no attempt made by the authors to define the HA head epitope bound by 651. General execution of this study was sufficient overall, but there were some unexplained results from the co-depletion and cytotoxicity assays that are not satisfactorily addressed.

**Part II – Major Issues: Key Experiments Required for Acceptance**

Reviewer #1: (No Response)

Reviewer #2: (No Response)

Reviewer #3: The authors have not addressed the below experimental concern to my satisfaction:

Why is there a massive decrease in survival for the control liposome + control antibody mice that received 651 (Figure 5A, 3rd panel)? In this group, only 10% of mice that got 651 survived compared to 100% of mice that received F10. Why would the control liposomes + control antibody have this effect on 651 treated mice? This finding is completely ignored in the Results/Discussion and needs to be addressed. It is very strange the control liposomes and control antibody would have this kind of impact on protection by 651 and suggests some kind of experimental issue (possibly with the liposomes).

Reply: We thank the reviewer for noting the unexpected effect of control liposome (Figure 5A) on 651 (1st and 3rd panel) or on F10 (1st panel) treatment. Although have not been formally demonstrated, we suspected that intranasal administration of liposome alone might absorb or affect the function of antibody. We have included this point in this revision (page 14, lines 235-237).

The authors did respond to this comment, but they did NOT address why in the 3rd panel of Figure 5A the liposome + control antibody treatment brought the survival of 651 mAb treated mice down to only 10% while 100% of F10 mAb treated mice survived. If this was a non-specific antibody-liposome interaction as the authors suggest, why wasn't a reduction in survival observed for the F10 treated group as well? I was hopeful that the authors would have either repeated this mouse experiment or provided some kind of logical explanation for this strange result. Since they have done neither of these things, it is difficult to decipher the results of this critical murine co-depletion assay.

The competition ELISA performed with a single HA-head mAb represented the absolute minimum but it was satisfactory, if barely.

**Part III – Minor Issues: Editorial and Data Presentation Modifications**

Reviewer #1: (No Response)

Reviewer #2: (No Response)

Reviewer #3: None

PLOS authors have the option to publish the peer review history of their article (what does this mean?). If published, this will include your full peer review and any attached files.

Reviewer #1: No

Reviewer #2: No

Reviewer #3: No

Figure Files:

Data Requirements:

Reproducibility:

References:

---

## [Editor Report · Decision Letter 2]

18 Jun 2021

Dear Dr. Lin,

We are pleased to inform you that your manuscript 'A non-neutralizing antibody broadly protects against influenza virus infection by engaging effector cells' has been provisionally accepted for publication in PLOS Pathogens.

Best regards,

Sabra L. Klein

Associate Editor

PLOS Pathogens

Carolina Lopez

Section Editor

PLOS Pathogens

Kasturi Haldar

Editor-in-Chief

PLOS Pathogens

orcid.org/0000-0001-5065-158X

Michael Malim

Editor-in-Chief

PLOS Pathogens

orcid.org/0000-0002-7699-2064
---

## [Editor Report · Acceptance letter]

21 Jul 2021

Dear Dr. Lin,

We are delighted to inform you that your manuscript, "A non-neutralizing antibody broadly protects against influenza virus infection by engaging effector cells," has been formally accepted for publication in PLOS Pathogens.

Best regards,

Kasturi Haldar

Editor-in-Chief

PLOS Pathogens

orcid.org/0000-0001-5065-158X

Michael Malim

Editor-in-Chief

PLOS Pathogens

orcid.org/0000-0002-7699-2064